materials science/green chemistry

graphene oxide, green synthesis, Ag/graphene-based nanomaterials, antibacterial

**Authors for correspondence:**
Haitao Ni
e-mail: htniok@163.com
Heshan Yang
e-mail: yheshan@cqwu.edu.cn

This article has been edited by the Royal Society of Chemistry, including the commissioning, peer review process and editorial aspects up to the point of acceptance.

# Rapid synthesis and characterization of silver-loaded graphene oxide nanomaterials and their antibacterial applications

Jiang Zhu[1], Haitao Ni[1], Chunyan Hu[2], Yuxiang Zhu[2], Jinxia Cai[2], Song Liu[2], Jie Gao[2], Heshan Yang[2] and Hongpan Liu[2]

[1]Chongqing Key Laboratory of Environmental Materials and Remediation Technologies, Chongqing University of Arts and Sciences, Yongchuan 402160, People's Republic of China
[2]College of Chemistry and Environmental Engineering, Chongqing University of Arts and Sciences, Yongchuan 402160, People's Republic of China

HN, 0000-0001-5299-0630

With the promising potential application of Ag/graphene-based nanomaterials in medicine and engineering materials, the large-scale production has attracted great interest of researchers on the basis of green synthesis. In this study, water-soluble silver/graphene oxide (Ag/GO) nanomaterials were synthesized under ultrasound-assisted conditions. The structural characteristics of Ag/GO were confirmed by Fourier transform infrared spectroscopy, X-ray diffraction, transmission electron microscopy, scanning electron microscopy and energy dispersion spectroscopy, respectively. The results showed the silver particles (AgNPs) obtained by reduction were attached to the surface of GO, and there was a strong interaction between AgNPs and GO. The antibacterial activity was primarily evaluated by the plate method and hole punching method. Antibacterial tests indicated that Ag/GO could inhibit the growth of Gram-negative and Gram-positive bacteria, special for the *Staphylococcus aureus*.

## 1. Introduction

Metal nanoparticles (MNPs) have attracted much attention owing to their superior characteristics including large specific surface area, high physical stability, strong electron transfer ability and high light absorption, retaining many applications such as

catalysts and antibacterial agents [1,2]. Various MNPs have been investigated, especially silver nanoparticles (AgNPs), which are unique compared to other MNPs because of their spectral antibacterial properties. Many researchers have investigated the release of silver ions ($Ag^+$) from AgNPs, and AgNPs bind to enzymes in the thiol group and interfere with the respiratory chain of microorganisms, leading to cell damage and oxidative stress. Because of these characteristics, AgNPs are increasingly used as a general solution for antibacterial activity [3]. However, AgNPs exhibit two main disadvantages at the nanometre level. On the one hand, the instability of AgNPs will cause the particles to agglomerate and reduce the antibacterial activity. On the other hand, AgNPs have a high toxic potential for human cells at high concentrations. AgNP-mediated cytotoxicity is attributed to the induction of reactive oxygen species (ROS) caused by the release of $Ag^+$. To improve the antibacterial properties of the material and minimize the adverse effects of AgNPs, it is very important to apply novel nanostructures as an alternative to decorative AgNPs.

Graphene oxide (GO) is a derivative of graphene, a two-dimensional monoatomic slab composed of $sp^2$-hybridized carbon atoms. A great number of functionalized active groups such as hydroxyl groups, epoxy groups and carboxyl groups on the surface of GO, not only enhance their hydrophilicity and biocompatibility, but also promote their interactions with other molecules or surface modification of polymers [4–6]. However, owing to strong plane-to-plane interactions, GO tends to accumulate layer by layer in aqueous solutions, which potentially limits its application. It is worth noting that if nanoparticles are coated on the surface of the GO layer, these nanoparticles (such as Ag, Au and Pt) will separate adjacent flakes and prevent their aggregation [7–9]. Meanwhile, the abundant functional groups on the surface of GO can act as nucleation centres or anchor positioning points, which limits their growth and improves the stability and dispersion of MNPs just like AgNPs on GO nanosheets. Moreover, GO nanosheets can be used not only as an important carrier material to modify AgNPs, but also as an important carrier to improve the antibacterial properties.

Some synthesis methods of Ag/graphene-based nanomaterials have been reported recently. Kellici *et al*. [1] showed that calixarene assisted rapid synthesis of Ag–graphene nanocomposites by continuous hydrothermal flow synthesis, but p-sulfonic acid calixarene was used as the surfactant stabilizer. Moghayedi *et al*. [2] synthesized an Ag-RGO nanocomposite by a facile one-pot method under microwave radiation, but silver nitrate ($AgNO_3$) was used in large usage and high cost. Ganguly *et al*. [4] prepared quasi-spherical AgNPs decorated reduced GO by a top-down method, nevertheless rigorous reaction conditions such as pH were employed. AlAqad *et al*. [10] reported the *in situ* reduction of decorated AgNPs on graphene by sodium borohydride. However, the nanomaterials have major environmental problems and health hazards, namely toxic reducing agents. Tian *et al*. prepared Ag/GO nanomaterials by heating a mixture of GO and $AgNO_3$ aqueous solution in the presence of sodium hydroxide and magnetic stirring at 80°C, but the reaction time was too long [8]. All the mentioned preparation methods have the problems of strict reaction conditions, complicated post-processing and are time-consuming. These methods and modes of operation are limited in many areas. Therefore, using controllable reaction parameters, shorter time and green synthesis methods, there must be a simple, non-toxic green process to synthesize products.

Sonochemical synthesis is an eco-friendly method that follows the concept of self-aggregation and generates nanoparticles through a top-down method [11]. Ultrasonic energy produces acoustic cavitation, resulting in the formation of bubbles/vesicles, nucleation/growth and collapse in the liquid environment [12,13]. The burst of bubbles will release powerful heat energy and pressure in a short time. According to the published reports, the ultrasonic method can produce a temperature of 5000 K, a pressure of 20 MPa and a high cooling rate of $10^{10} \, K \, s^{-1}$ [4]. Through ultrasonic treatment, acoustic cavitation provides an effective environment for the reaction without using any strict reducing agent. In other words, the sonochemical method confirmed the principle of green chemistry. The development and application of new nanomaterials whose characteristics can combine MNPs with graphene or its derivatives have aroused much interest. To our knowledge, there are several ways to incorporate MNPs into graphene and its derivatives, such as chemical methods, chemical vapour deposition and wave radiation [7,11]. Among them, ultrasonic radiation from $Ag^+$/GO to Ag/GO is the most suitable and effective method. Ultrasonic radiation can provide local voids, which can increase temperature and pressure in an instant, resulting in the acceleration of chemical reactions for the synthesis of various nanomaterials. Furthermore, ultrasonic radiation can form nanoparticles of uniform size and shape in a short reaction time.

On account of the above reasons, a simple method to reduce $Ag^+$ on GO sheets at room temperature using ascorbic acid (AA) as a reducing agent was proposed in this study. Because AA has moderate reducing power and non-toxic properties. It can be used as a reducing agent in natural organisms and

the main reducing agent in the laboratory [14]. Compared with conventional reducing agents for $Ag^+$ reduction (such as hydrazine and hydrazine hydrate), both AA itself and its oxidation products are environmentally friendly. We were dedicated to the green synthesis of high-quality Ag/GO nanomaterials by ultrasonic radiation reduction to achieve the loading of AgNPs on GO nanomaterials at ambient temperature. The method of producing Ag/GO is easy to operate and the reaction time is short, thus further opening up a new field of mass production of graphene. In addition, the structure of the final product was also investigated by Fourier transform infrared (FTIR) spectroscopy, X-ray diffraction (XRD), transmission electron microscopy, scanning electron microscopy and energy dispersion spectroscopy (EDS), respectively. Also, the antibacterial activity was primarily evaluated by the plate method and hole punching method.

# 2. Material and methods

## 2.1. Materials

Graphite powders (325 mesh) with a 99.95% purity were obtained from Macklin. AA with a 99% purity was obtained from Aladdin. Silver nitrate ($AgNO_3$, AR) was obtained from the Chengdu Cologne Chemical Co., Ltd. Chemical agents including 98% sulfuric acid ($H_2SO_4$) and 38% hydrochloric acid (HCl) were produced from the Chongqing Chuandong Chemical Co., Ltd.. Potassium permanganate ($KMnO_4$) was purchased from the Aladdin Chongqing Beibei Chemical Reagent Factory. Sodium hydroxide (NaOH), 30% hydrogen peroxide ($H_2O_2$), which were manufactured from the Chengdu Kelong Chemical Reagent Factory. Deionized water was used to prepare all the solutions. The materials were used directly without any further purification.

## 2.2. Synthesis of silver/graphene oxide nanomaterials

### 2.2.1. Preparation of graphene oxide

Graphite oxide was prepared by a modified Hummers' method [15,16]. Graphite oxide was prepared by oxidizing graphite powder with concentrated $H_2SO_4$ and $KMnO_4$ without adding sodium nitrate. After purifying, the graphite oxide was ultrasonically exfoliated to produce GO. After further purification of the GO, it was freeze dried for later use.

### 2.2.2. Fabrication of silver-loaded reduced graphene oxide

Ag/GO nanomaterials were prepared by mixing $AgNO_3$ water solution at various concentrations (0.2, 0.4 and 0.8 mM), and 250 ml of GO suspensions (0, 0.5, 1 mg ml$^{-1}$) under a magnetic stirring for 30 min. Afterward, AA and NaOH (the molar ratio of $AgNO_3$, AA and NaOH is 1 : 1 : 1.5) were slowly added into the reaction mixture. The mixture was treated under ultrasound (SCIENTZ-1200E, 1200 W, 25 kHz) conditions. Table 1 shows the different reaction conditions in the orthogonal experiment, based on the previous literature [13].

### 2.2.3. Antibacterial test

The antibacterial activity of Ag/GO nanomaterials against Gram-negative bacteria *Escherichia coli* ATCC 25922 and Gram-positive bacteria *Staphylococcus aureus* ATCC 6538 was studied by two methods: plate method and hole punching method. *Escherichia coli* cells and *S. aureus* were cultured in a sterile clean room [17,18]. All the used equipment was sterilized in autoclave under high-pressure and high-temperature conditions. The strain was incubated in 5 ml tryptone soy broth (TSB; Oxoid, UK). Bacterial cultures were grown in an incubator-shaker at 310 K. After shaking for 24 h, they were diluted to $1 \times 10^6$ CFU ml$^{-1}$ of the bacterial suspension. The diluted bacterial suspension solution was then applied to tryptone soy agar (Oxoid, UK) plates using a triangular applicator. A 120 µl sample solution was added to a 6 mm well. The inhibition zone was then examined by incubating in a biochemical incubator for 24 h at 310 K. A bacterial suspension (500 µl) was inoculated into 5 ml of TSB containing the sample solution (500 µl) and statically cultured overnight at 310 K. Then, the above solution (100 µl) was taken on a plate. Spectrophotometric absorbance was measured at a wavelength of 590 nm in a microtiter plate reader (Cytation 5, Biotek, USA). The growth medium to which phosphate buffered saline solution was added was used as a negative control. The cell viability

**Table 1.** Ultrasonic reaction orthogonal table.

| number | GO (mg ml$^{-1}$) | AgNO$_3$ (mM) | ultrasound time (min) | ultrasound power (W) |
|---|---|---|---|---|
| 1 | 0 | 0.2 | 10 | 400 |
| 2 | 0 | 0.4 | 20 | 800 |
| 3 | 0 | 0.8 | 30 | 1200 |
| 4 | 0.5 | 0.2 | 20 | 1200 |
| 5 | 0.5 | 0.4 | 30 | 400 |
| 6 | 0.5 | 0.8 | 10 | 800 |
| 7 | 1 | 0.2 | 30 | 800 |
| 8 | 1 | 0.4 | 10 | 1200 |
| 9 | 1 | 0.8 | 20 | 400 |

was calculated by: cell viability $(100\%) = OD_t/OD_n \times 100\%$, where $OD_t$ and $OD_n$ are the absorbance of the test and the absorbance of negative control, respectively.

## 2.3. Characterization

Absorption spectroscopy was obtained on a UV-500PC spectrophotometer (Shanghai Metash, China). The 200–800 nm range and 1 nm resolution were used to analyse the solution prepared in the orthogonal experiment.

The group structure analysis was performed at room temperature on a FTIR spectrometer (Nicolet T6670; Thermo Fisher Scientific, America). The scanning range was 4000–400 cm$^{-1}$. A sample of 1 mg was ground with 200 mg of dried potassium bromide (KBr, 99.5%, Merck) to prepare a test sample of FTIR. The KBr was dried at 60°C for more than 24 h.

The zeta potential of the particles in the Ag/GO solution was detected by a JS94K2 type microelectrophoresis apparatus (Shanghai Zhongchen Digital Technology Equipment Co., Ltd., China). The stability of the Ag/GO colloidal solution was analysed.

XRD pattern of the Ag-RGO nanomaterials was obtained on a DX-1000 diffractometer (Dandong Fang-yuan Instrument Co., Ltd., China) using CuK$_\alpha$ radiation ($\lambda = 1.50411$ Å). The scanning speed was $6°$ min$^{-1}$ and the scanning range was from 10° to 80°. The voltage and current were set to 35 kV and 20 mA, respectively.

The surface morphologies were performed on a scanning electron microscope (ZEISS Gemini-SEM 300, Germany) with EDS. To observe the morphology of the AgNPs more clearly, there was no gold spray.

The high-resolution transmission electron microscopy (HRTEM) images were taken by a JEM-2100 model instrument operated at an accelerating voltage of 100 kV. Samples for HRTEM imaging were prepared by placing a drop of the sample solution in a carbon-coated copper grid and then dried in air.

The X-ray photoelectron spectra of Ag/GO were obtained with an X-ray photoelectron spectrometer (XPS; Thermo Scientific) using a flood gun charge neutralizer system equipped with an Al K-α X-ray source ($h\upsilon = 1486.6$ eV).

# 3. Results and discussion

## 3.1. Preparation of silver/graphene oxide

Figure 1 schematically displays the synthesis of Ag/GO nanomaterials. After treating by H$_2$SO$_4$, the surface of GO contained many oxygen-containing functional groups, such as epoxy, hydroxyl, carbonyl and carboxylic acid groups. These functional groups can be used as negative active sites for metal cations [10]. So the surface of GO adsorbed Ag$^+$ to form Ag$^+$-GO precursors in the solution. When ultrasound was added, ultrasonic vibration can increase the movement of Ag$^+$-GO precursors owing to the mechanical effect and thermal effect of ultrasound. Therefore, under assisting with

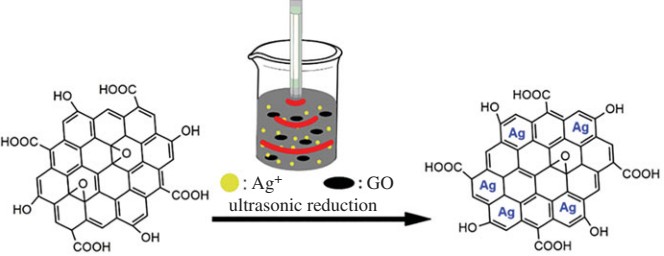

**Figure 1.** Schematic diagram of the synthesis of Ag/GO nanomaterial.

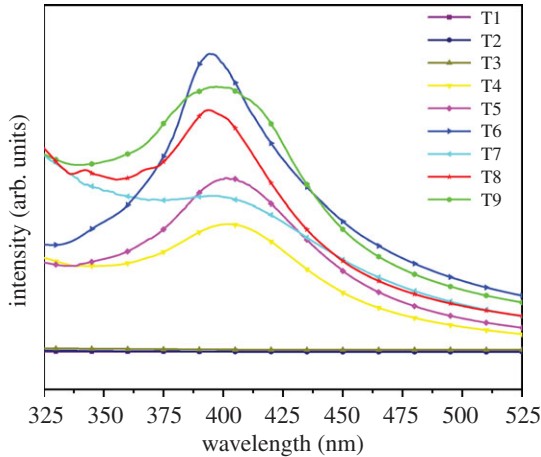

**Figure 2.** UV-visible spectrum of Ag/GO nanomaterials (T1–T9).

ascorbic acid as reducing agent, the Ag⁺ on the surface of GO could be reduced to silver particles and Ag/GO nanomaterials are finally formed.

## 3.2. Structural analysis of silver/graphene oxide

Figure 2 shows the ultraviolet(UV)-visible light absorption spectrum of water-dispersed Ag/GO nanomaterials. T4–T9 samples had an obvious characteristic absorption peak at approximately 400 nm [2]. Such a peak was attributed to the characteristic of the surface plasmon resonance band of AgNPs, indicating that AgNPs presented in the synthesized product [18]. However, there was no characteristic peak in the T1–T3 samples, which could be attributed to the accumulation and sedimentation of the generated AgNPs. These results indicated that the presence of GO had a certain effect on the stable dispersion of AgNPs. Meanwhile, according to the absorption intensity, it can be seen that the absorption intensity of the T6 sample was highest among all samples. Therefore, T6 was selected as the representative sample.

XRD was used to characterize the crystalline phase of Ag/GO nanomaterials. From the XRD results in figure 3, Ag/GO nanomaterials had clear diffraction peaks at $2\theta = 38, 44, 64$ and $77°$, corresponding to (111), (200), (220) and (311) crystal planes, respectively [19,20]. These data are consistent with the AgNPs standard card (JCPDS No. 04-0783) with a face-centred cubic structure. Similarly, the existence of all diffraction peaks of AgNPs also confirmed that the phase and crystallinity in the form of nanomaterials were retained. Unexpectedly, after modifying GO with AgNPs, the strong peak of GO disappeared at $10.7°$ [21]. The reason for this phenomenon could be that GO was partially reduced, reducing the oxygen-containing functional groups, thereby shortening the distance between layers.

Figure 4 exhibits the FTIR spectra of graphite powder, GO and Ag/GO nanomaterials. Compared with GO, the peaks at approximately 1628 and 1395 cm⁻¹ were ascribed to skeleton vibrations of aromatic C=C bonds [22]. The frequency bands at approximately 3130 and 1120 cm⁻¹ were attributed to C-H tensile vibration and C–O tensile vibration, respectively [19]. The broad and strong peaks related to the –OH group were concentrated at approximately 3433 cm⁻¹. The peak at approximately 1728 cm⁻¹ for GO sample corresponded to the tensile vibration peak of the C=O carboxyl moiety [23].

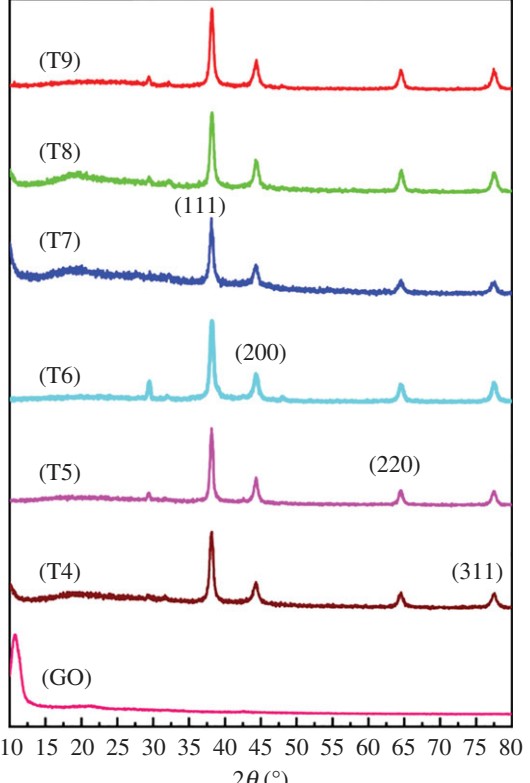

**Figure 3.** XRD of GO and Ag/GO nanomaterials (T4–T9).

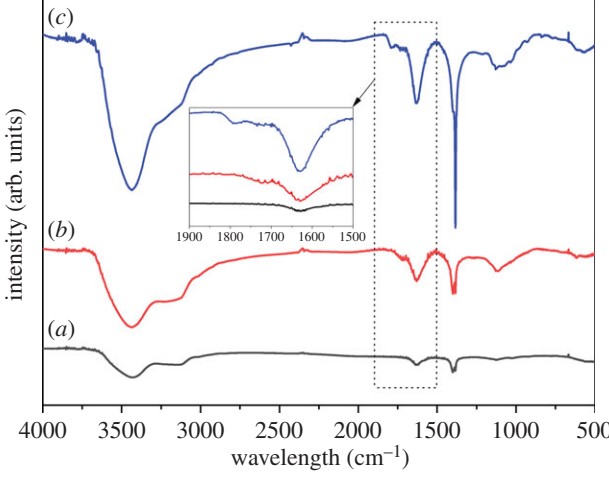

**Figure 4.** FTIR spectra of (*a*) graphite, (*b*) GO and (*c*) Ag/GO (T6) nanomaterials.

Thus, it can be proved that graphite powder had been successfully oxidized and the surface oxidation of GO produced many oxygen-containing groups, forming many active sites on GO. For Ag/GO nanomaterials, there were still positions of functional group peaks on GO, but the C=O carbonyl tensile strength (approx. 1728 cm$^{-1}$) decreased and flexure occurred, while aromatic C=C vibration (approx. 1395 cm$^{-1}$) belonged to the GO nanosheets. It became very sharp and strong [24,25]. This transition can be confirmed the formation of chemical bonds or electrostatic attraction between silver and GO nanoparticles, which was consistent with similar reports [6,22].

Figure 5 demonstrates the XPS results of GO and Ag/GO samples. The surface compositions were determined from these spectra. Three conjunct peaks are observed in figure 5c. One peak located at approximately 284.8 eV was assigned to C–C/C=C, and another two peaks at approximately 287.1 eV and 289.0 eV were ascribed to C–O and C=O bonds, respectively [26,27]. Simultaneously, the similar

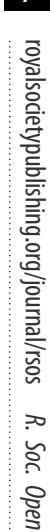

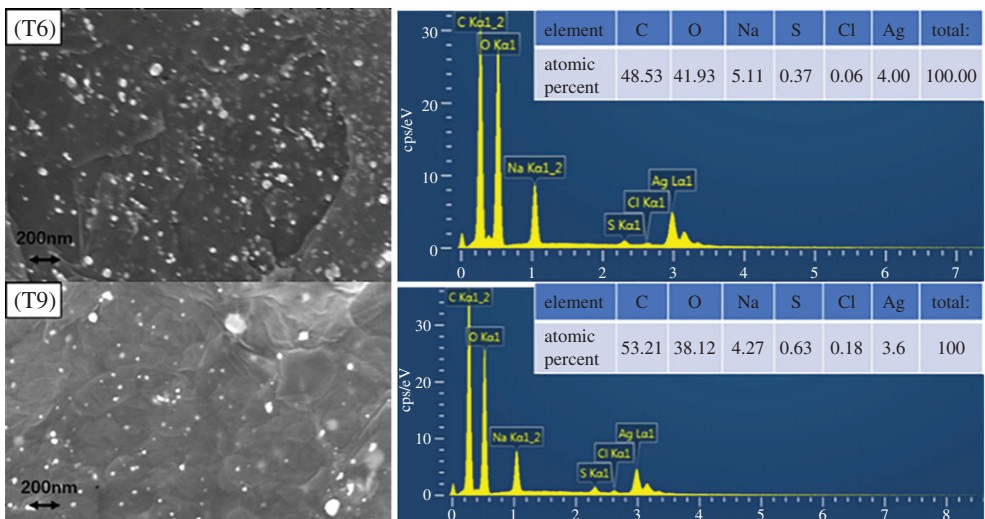

**Figure 5.** (*a*) Survey XPS spectrum of GO and Ag/GO, (*b*) high-resolution of XPS spectrum of Ag 3d of Ag/GO samples, (*c*) C 1 s XPS profile for GO, and (*d*) Ag/GO-T6 nanomaterials.

**Figure 6.** Field emission scanning electron microscopy images of Ag/GO and the EDS curve for the selective T6 and T9 sample.

typical peaks were also observed in the Ag/GO nanomaterials. The appearance of the double peak of Ag 3d was confirmed by the deposition of AgNPs on GO. As shown in figure 5*b*, the Ag 3d high-resolution spectrum consisted of two different peaks. Both spectrum showed the Ag 3d level of spin-orbit split, namely in $E = 373 \sim 376$ eV binding energy ($E$) of Ag-3d$_{3/2}$ and Ag-3d$_{5/2}$ in $E = 367$–$370$ eV spin-orbit

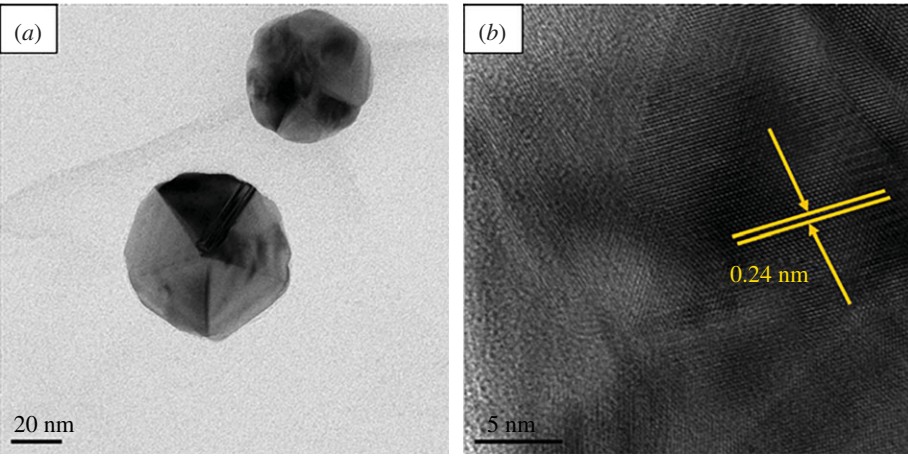

**Figure 7.** HRTEM image of Ag/GO nanomaterials (T6).

electron volts, which indicated the presence of Ag in the prepared samples. Owing to the spin coupling $\Delta \hat{E} = 6.0$ eV, it clearly indicated the existence of Ag$^0$ [28]. These peaks located at lower binding energies should not be appointed to oxide silver compounds, because Ag 3d in oxides was located at more powerful binding energies, e.g. Ag-3d$_{5/2}$ = 367.7 eV for Ag$_2$O and 367.4 eV for that of AgO [29]. These peaks were attributed to the structure of Ag–C$_{5/2}$ (approx. 369 eV) and Ag–C$_{3/2}$ (approx. 375 eV) components [30]. Furthermore, the peaks had strong interactions between silver and carbon on the benzene ring, which was consistent with the results of FTIR.

## 3.3. Morphology analysis of silver/graphene oxide

Figure 6 displays the surface morphology and the elemental composition of the T6 and T9 sample, respectively. The selective samples had a lot of uneven spots on the surface, randomly covering the surface of GO nanosheet. These points were attributed to spherical nanoparticles of AgNPs [31]. The EDS data for T6 and T9 samples show that the Ag/GO nanomaterials contained elements such as carbon (C), oxygen (O) and silver (Ag). The presence of a small amount of sulfur and chlorine elements was owing to impurities that were introduced by using H$_2$SO$_4$ and HCl in the preparation of GO. Nevertheless, the presence of Ag, C and O confirmed the formation of the Ag/GO nanomaterials. GO had a large surface area and contained a large amount of oxygen-containing functional groups. These groups on the GO sheet provided the chemically active centres for the deposition of AgNPs, while the AgNPs were well separated from each other [7]. In addition, it was noteworthy that the C:O ratio in the T6 sample was less than that of the T9 sample, but the Ag content was higher than T9, mainly owing to the fact that the oxygen-containing groups on the GO sheet were the main active points in the reaction, which could attract Ag$^+$ onto the GO surface [22].

Figure 7 shown the HRTEM results of the Ag/GO nanomaterial (T6 sample). The deposited AgNPs was an irregular sphere and had an average diameter of approximately 45 nm, seen in figure 7a. It was indicated that GO nanosheets played an important role in the nucleation and stabilization in the preparation process for AgNPs. GO nanosheets acted as the morphological drivers for AgNPs, which determined the formation of Ag spherical particles in Ag/GO nanomaterials. From the image with higher magnification(figure 7b), it can be found that AgNPs were embedded on the surface of the GO nanosheet and exhibited a variety of structures. Additionally, the measured crystal lattice of AgNPs was 0.24 nm, corresponding to the (111) crystal plane [4,32]. These results were also consistent with both XPS and XRD results.

## 3.4. Performance evaluation of silver/graphene oxide

Figure 8 exhibited zeta potentials of GO and Ag/GO nanomaterials. Both the GO (−83.91 ± 0.23 mV) and Ag/GO (−44–84 mV) presented negative charges owing to the negative-charged functional groups (such as C–O–C, −COOH and −OH) on the surface of GO. It was well known that the colloid would have excellent stability when the absolute value of the zeta potential was greater than 30 mV [33]. It can be seen that although the zeta potential values of the T4–T6 samples were slightly lower than the

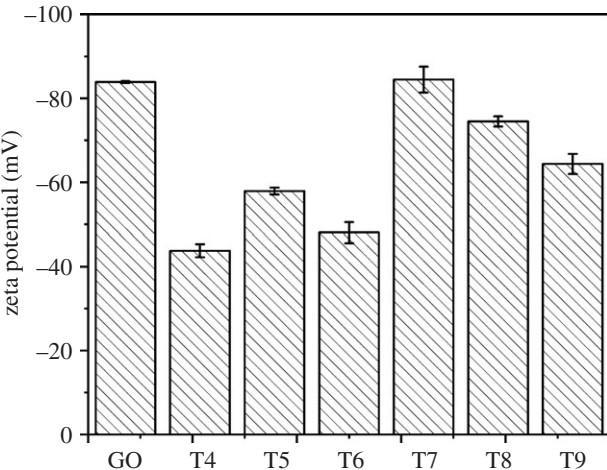

**Figure 8.** Zeta potential of GO and Ag/GO nanomaterials.

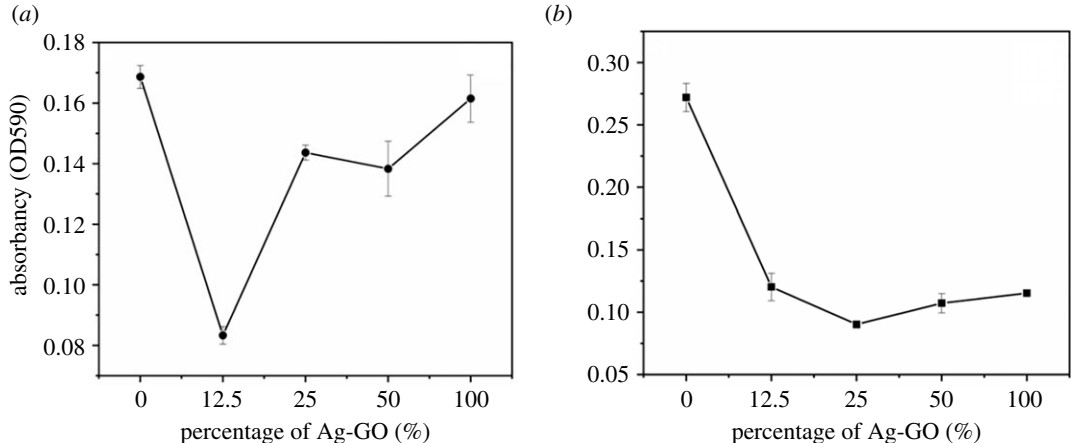

**Figure 9.** Absorbance of *E. coli* (*a*) and *S. aureus* (*b*) after treatments with Ag/GO nanomaterials (T6).

counterpart of the T7–T9 samples, the absolute value of the T6 sample (approx. 48 mV) was 60% higher than the standard value of 30 mV. It indicated clearly that the selective Ag/GO nanomaterials had good stability in aqueous solution. This result could be explained by the fact that the existence of negative-charged functional groups on the surface of GO could produce the greater electrostatic repulsion between Ag/GO nanomaterials and would be a benefit for preventing the agglomeration of Ag/GO nanomaterials, resulting in a more stable dispersion of nanoparticles.

Both Gram-negative *E. coli* and Gram-positive *S. aureus* were selected for the primary antimicrobial testing, which is commonly associated with medically relevant infections. The OD values were tested using Cytation 5, and the effect of the content of the selective Ag/GO sample (T6) on bacterial cell viability was investigated. From figure 9, it can be seen that the increment of the Ag/GO content had a different effect on the absorbance of the treated medium. For example, when the percentage of the Ag/GO sample increased from 12.5% to 100%, the absorbance of the *E. coli* medium decreased first and then increased [34]. However, for the *S. aureus* medium, the corresponding values decreased rapidly at the 12.5% percentage of Ag/GO and were inclined to be placid beyond 12.5%. Based on the above data, the cell viability of the two bacteria was obtained by the calculation of the absorbance of the test medium against the absorbance of negative control, respectively. The related data are shown in figure 10. In the cell viability assay of *S. aureus*, as the concentration of the Ag/GO specimen increased from 12.5% to 100% the cell viability decreased gradually to be stable. Nevertheless, it was noticeable that the cell viability in the *E. coli* medium decreased first and then increased when the concentration of Ag/GO was beyond 12.5%. It is well known that adding content of an antimicrobial agent has an important effect on the cell viability. For the metal ions antibacterial agents, metal ions with the positive charge can be adsorbed on the surface of the bacterial cell membrane with the

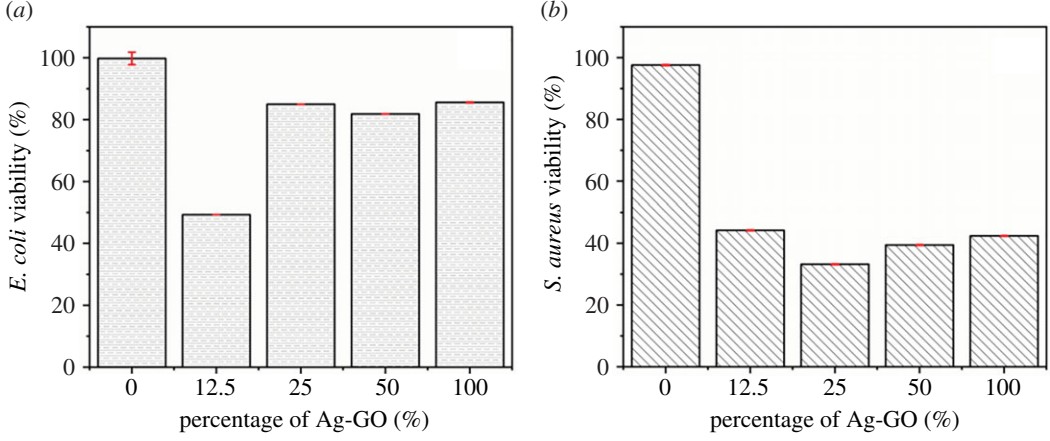

**Figure 10.** Bacterial cell viability of (*a*) *E. coli* and (*b*) *S. aureus* after treatments with Ag/GO nanomaterials.

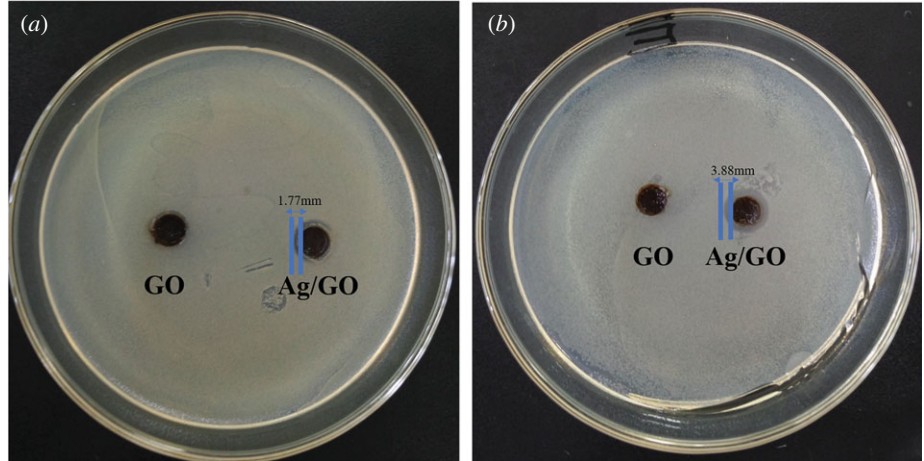

**Figure 11.** GO and Ag/GO sample (T6) inhibition zone size for (*a*) *E. coli* and (*b*) *S. aureus*.

negative charge to prevent cell movement and reduce cell activity. According to the results, it can be concluded that Ag/GO nanomaterials presented negative charges (−44−84 mV) owing to the negatively charged functional groups on the surface of GO. In the cell viability assays, the charges on the surface of *E. coli* bacteria were higher than the counterparts of *S. aureus* bacteria. So when the concentration of the Ag/GO sample with negative charges increased, the repulsive forces between Ag/GO nanomaterials with the selective bacteria increased, especially for that of *E. coli*. It may prevent the release of $Ag^+$ migrating from the surface of the Ag/GO nanomaterial to the *E. coli* cell membrane. Furthermore, the cell membrane of *E. coli* was thicker than that of *S. aureus*. It could hinder the $Ag^+$ permeating into the cell membrane of *E. coli* at the same time. These results may be attributed to the different sensitivies of Ag/GO nanomaterials to different bacteria. Moreover, the pH values could also affect the cell viability in the culture solution with the different percentages of the Ag/GO nanomaterials, resulting in a limited amount of $Ag^+$ release. Further investigation in detail are in progress and will be reported in the follow-up work.

Figure 11 displays the inhibition zone tests of GO and Ag/GO nanomaterial (T6) against *E. coli* and *S. aureus* bacteria. One can be seen that there was a clear inhibition zone for Ag/GO nanomaterial in both *E. coli* and *S. aureus* medium, compared with the controlled experiment of GO. Moreover, the inhibition zone for *S. aureus* (approx. 3.88 mm) was almost twice the size of *E. coli* (approx. 1.77 mm). These results were identical with the data of bacterial cell viability for *E. coli* and *S. aureus*. It manifested that the introduction of AgNPs onto the surface of GO improved the antimicrobial effects of pure GO. The possible mechanism was owing to $Ag^+$ released from AgNPs could strongly bind with proteins on the surface of the cell membrane and further disturb the multiplication of cells, leading to the extinction of bacteria [2,3].

# 4. Conclusion

In conclusion, an environmentally friendly, simple and fast method was developed to synthesize Ag/GO nanomaterials under ultrasound-assisted conditions. The infrared and XPS indicated that there was a strong interaction between AgNPs and GO, so that AgNPs are well dispersed on the surface of the GO nanosheet. The zeta potential analysis shows that the suspension of Ag/GO nanomaterials was stable enough to withstand the accumulation of nanomaterials. In addition, the Ag/GO nanomaterials exhibited an antibacterial activity against *E. coli* and *S. aureus*. The results indicate that Ag/GO nanomaterials may be a promising antibacterial agent, broadening the scope of application for Ag/GO nanomaterials.

Data accessibility. All data relevant to this work are deposited at the Dryad Data Repository [35]: (https://doi.org/10.5061/dryad.zgmsbcc8r).

Authors' contributions. J.Z., H.N. and H.Y. designed the study. C.H. and Y. Z. prepared all nanomaterial samples. J.C., S.L., J.G and H.L. collected and analysed the data. J.Z., H.N. and H.Y. interpreted the results and wrote the manuscript. All authors gave final approval for publication.

Competing interests. We have no competing interests.

Funding. This research was funded by the Science and Technology Innovation Project for Social Undertaking and Livelihood System of Chongqing (grant no. cstc2017shmsA30018), the Science and Technology Project of Chongqing City Administration Bureau (grant no. 2019-26) and the Natural Science Foundation Projects of Chongqing (grant no. cstc2020jcyj-msxmX0632).

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
