## [Peer Review File · Royal Society Open Science]

Review History

RSOS-201744.R0 (Original submission)

Review form: Reviewer 1

Is the manuscript scientifically sound in its present form?

Yes

Are the interpretations and conclusions justified by the results?

No

Is the language acceptable?

Yes

Do you have any ethical concerns with this paper?

No

Have you any concerns about statistical analyses in this paper?

Yes

Recommendation?

Major revision is needed (please make suggestions in comments)

Comments to the Author(s)

In this report, the authors reported the preparation of Ag/GO nanomaterial and its applications as antimicrobial agents. This manuscript needs a major revision and rereview. The comments are given below.

1. The review of literature suggests that previous successful attempts have been made to develop the Ag nanomaterial as well as its antimicrobial activity so pertaining to this novelty of this investigation has to be clarified in more detail.

Following references about different novel Ag/GO bases nanomaterials developed with their multifunctional properties are mentioned ACS Appl. Mater. Inter. 8, 19038-19046. (doi:10.1021/acsami.6b06052), Colloids Surface B 159, 366-374., Coordin. Chem. Rev. 357, 1-17. Ultrason Sonochem. 39, 577-588.

2. In the preparation process of Ag/GO, on what basis, the amount of reagents ultrasound time and ultrasound power were determined? Were the high combination of GO and Ag tried with 10min, 30min for ultra sound synthesis.

3. The growth mechanism for the formation of nanomaterial Ag/GO should be explained in more detail based on the chemistry concept.

4. In the Bacterial Cell viability assay, as the concentration of nanomaterial is increased from 12.5% - 100% the cell viability is increasing, this is contradictory to it being applied as an antimicrobial agent. Kindly Give a detailed justification.

5. For an antimicrobial agent the minimum concentration of substance should be effective. So the calculations pertaining to be MIC has to be done. As high concentration would be toxic to other living cells.

Review form: Reviewer 2**Is the manuscript scientifically sound in its present form?**

Yes

Are the interpretations and conclusions justified by the results?

Yes

Is the language acceptable?

Yes

Do you have any ethical concerns with this paper?

No

Have you any concerns about statistical analyses in this paper?

No

Recommendation?

Accept with minor revision (please list in comments)

Comments to the Author(s)

In this work, the synthesis and characterization of silver-loaded graphene oxide nanomaterials have been systematically investigated. The antibacterial test has been also conducted by two methods. I recommend this paper for publication after minor revision. Here listed some issues should be clarified.

1. the abstract is hard to read and master what the authors want to demonstrate? Please reorganization to make the manuscript to understand.
2. The literatures in the introduction section can not well cover the research progress of Graphene oxide, Metal nanoparticles, Sonochemical synthesis and so on. please add some related paper in the revision.
3. Figure 4 is used to prove graphite powder has been successfully oxidized and GO surface oxidation produces many oxygen-containing groups. However, the absorption peak at $\sim 1728\text{cm}^{-1}$ is too small to observe, please magnify this peaks.
4. Diameter of bacteriostatic zone in Figure 11 should be measured.
5. Authors should carefully check the grammar throughout the manuscript.
6. Is there any effect of ascorbic acid concentration and pH of the solution on the size and aggregation of AgNPs on GO?
7. Why did the author choose ultrasound assisted AgNPs synthesis instead of microwave-assisted in the experiment?
8. What is the effect of Ag particle size on antibacterial activity?

Decision letter (RSOS-201744.R0)

Dear Dr Ni:

Title: Rapid synthesis and characterization of silver-loaded graphene oxide nanomaterials and their antibacterial applications

Manuscript ID: RSOS-201744

The editor assigned to your manuscript has now received comments from reviewers. We would like you to revise your paper in accordance with the referee and Subject Editor suggestions which can be found below (not including confidential reports to the Editor). Please note this decision does not guarantee eventual acceptance.

Please submit your revised paper before 25-Dec-2020. Please note that the revision deadline will expire at 00.00am on this date. If we do not hear from you within this time then it will be assumed that the paper has been withdrawn. In exceptional circumstances, extensions may be possible if agreed with the Editorial Office in advance. We do not allow multiple rounds of revision so we urge you to make every effort to fully address all of the comments at this stage. If

deemed necessary by the Editors, your manuscript will be sent back to one or more of the original reviewers for assessment. If the original reviewers are not available we may invite new reviewers.

On behalf of the Subject Editor Professor Anthony Stace and the Associate Editor Dr Dattatray Late.

RSC Associate Editor:
Comments to the Author:
Major Revision needed

RSC Subject Editor:
Comments to the Author:
(There are no comments.)

Reviewers' Comments to Author:
Reviewer: 1

Comments to the Author(s)
In this report, the authors reported the preparation of Ag/GO nanomaterial and its applications as antimicrobial agents. This manuscript needs a major revision and rereview. The comments are given below.

1. The review of literature suggests that previous successful attempts have been made to develop the Ag nanomaterial as well as its antimicrobial activity so pertaining to this novelty of this investigation has to be clarified in more detail.

Following references about different novel Ag/GO bases nanomaterials developed with their multifunctional properties are mentioned ACS Appl. Mater. Inter. 8, 19038-19046. (doi:10.1021/acsami.6b06052), Colloids Surface B 159, 366-374., Coordin. Chem. Rev. 357, 1-17. Ultrason Sonochem. 39, 577-588.

2. In the preparation process of Ag/GO, on what basis, the amount of reagents ultrasound time and ultrasound power were determined? Were the high combination of GO and Ag tried with 10min, 30min for ultra sound synthesis.

3. The growth mechanism for the formation of nanomaterial Ag/GO should be explained in more detail based on the chemistry concept.

4. In the Bacterial Cell viability assay, as the concentration of nanomaterial is increased from 12.5% - 100% the cell viability is increasing, this is contradictory to it being applied as an antimicrobial agent. Kindly Give a detailed justification.

5. For an antimicrobial agent the minimum concentration of substance should be effective. So the calculations pertaining to be MIC has to be done. As high concentration would be toxic to other living cells.

Reviewer: 2

Comments to the Author(s)

In this work, the synthesis and characterization of silver-loaded graphene oxide nanomaterials have been systematically investigated. The antibacterial test has been also conducted by two methods. I recommend this paper for publication after minor revision. Here listed some issues should be clarified.

1. the abstract is hard to read and master what the authors want to demonstrate? Please reorganization to make the manuscript to understand.
2. The literatures in the introduction section can not well cover the research progress of Graphene oxide, Metal nanoparticles, Sonochemical synthesis and so on. please add some related paper in the revision.
3. Figure 4 is used to prove graphite powder has been successfully oxidized and GO surface oxidation produces many oxygen-containing groups. However, the absorption peak at $\sim 1728\text{cm}^{-1}$ is too small to observe, please magnify this peaks.
4. Diameter of bacteriostatic zone in Figure 11 should be measured.
5. Authors should carefully check the grammar throughout the manuscript.
6. Is there any effect of ascorbic acid concentration and pH of the solution on the size and aggregation of AgNPs on GO?
7. Why did the author choose ultrasound assisted AgNPs synthesis instead of microwave-assisted in the experiment?
8. What is the effect of Ag particle size on antibacterial activity?

Author's Response to Decision Letter for (RSOS-201744.R0)

See Appendix A.

Decision letter (RSOS-201744.R1)

Dear Dr Ni:

Title: Rapid synthesis and characterization of silver-loaded graphene oxide nanomaterials and their antibacterial applications
Manuscript ID: RSOS-201744.R1

It is a pleasure to accept your manuscript in its current form for publication in Royal Society Open Science. The chemistry content of Royal Society Open Science is published in collaboration with the Royal Society of Chemistry.

On behalf of the Subject Editor Professor Anthony Stace and the Associate Editor Dr Dattatray Late.

RSC Associate Editor
Comments to the Author:
Authors have revised the manuscript as per reviewer's suggestions.

Reviewer(s)' Comments to Author:

Appendix A

Dear Editor/Reviewers:

First of all, we would like to thank the reviewers of our paper (RSOS-201744) for his/her constructive and informative comments and suggestions. We have carefully considered the reviewer's comments and suggestions and have modified our manuscript accordingly. The following is our detailed response to the reviewer's comments and suggestions. For clearance, we have listed the reviewer's comments below and have addressed them one by one.

Reviewer Comments:

Reviewer: 1

Comments to the Author(s)

In this report, the authors reported the preparation of Ag/GO nanomaterial and its applications as antimicrobial agents. This manuscript needs a major revision and rereview. The comments are given below.

1. The review of literature suggests that previous successful attempts have been made to develop the Ag nanomaterial as well as its antimicrobial activity so pertaining to this novelty of this investigation has to be clarified in more detail.

Following references about different novel Ag/GO bases nanomaterials developed with their multifunctional properties are mentioned ACS Appl. Mater. Inter. 8, 19038-19046. (doi:10.1021/acsami.6b06052), Colloids Surface B 159, 366-374., Coordin. Chem. Rev. 357, 1-17. Ultrason Sonochem. 39, 577-588.

Our response: It is a good suggestion. The main purpose of this paper is to synthesize Ag/GO nanomaterials through the green method. And the ultrasound-assisted synthesis offers exceptional advantages over other approaches such as surfactant free, low temperature and so on. More detailed comparison or depiction has been clarified in the revised manuscript.

2. In the preparation process of Ag/GO, on what basis, the amount of reagents ultrasound time and ultrasound power were determined? Were the high combination of GO and Ag tried with 10min, 30min for ultrasound synthesis.

Our response: In the preparation process of Ag/GO, the basic reaction conditions were determined by the reference (Acar Bozkurt, P. 2017. "Sonochemical Green Synthesis of Ag/Graphene Nanocomposite." Ultrason Sonochem 35: 397-404). And the longer ultrasonic time selected in this paper would be beneficial to the formation of Ag nanoparticles on graphene surface and the dispersion of Ag/GO nanocomposites. The SEM (as following picture shows) showed that the silver ions can be transformed into silver nanoparticles and deposited on the surface of graphene oxide within the set time.

3. The growth mechanism for the formation of nanomaterial Ag/GO should be explained in more detail based on the chemistry concept.

Our response: The growth mechanism about the formation of nanomaterial Ag/GO has been explained in this paper again, based on the chemistry concept.

4. In the Bacterial Cell viability assay, as the concentration of nanomaterial is increased from 12.5% - 100% the cell viability is increasing, this is contradictory to it being applied as an antimicrobial agent. Kindly Give a detailed justification.

Our response: The detailed explanation has been re-described in the corresponding part in article.

5. For an antimicrobial agent the minimum concentration of substance should be effective. So the calculations pertaining to be MIC has to be done. As high concentration would be toxic to other living cells.

Our response: The purpose of this experiment is to prepare Ag/graphene-based nanomaterials through the green and convenient way by ultrasonic radiation. And the prepared Ag/GO nanomaterials will be a good precursor material that could be used not only in antibacterial fields but also in electrochemical fields. Therefore, in this work we conducted primary performance evaluation of Ag/GO on the antibacterial activity. For an antimicrobial agent the minimum concentration of substance is exactly effective. So the relevant specific works about the calculations pertaining to be MIC is being carried out and will be reported in the follow-up work in detail.

Reviewer: 2

Comments to the Author(s)

In this work, the synthesis and characterization of silver-loaded graphene oxide nanomaterials have been systematically investigated. The antibacterial test has been also conducted by two methods. I recommend this paper for publication after minor revision. Here listed some issues should be clarified.

1. the abstract is hard to read and master what the authors want to demonstrate? Please reorganization to make the manuscript to understand.

Our response: we have reorganized the abstract.

2. The literatures in the introduction section can not well cover the research progress of Graphene oxide, Metal nanoparticles, Sonochemical synthesis and so on. please add some related paper in the revision.

Our response: Some related paper has been added in the revised manuscript.

3. Figure 4 is used to prove graphite powder has been successfully oxidized and GO surface oxidation produces many oxygen-containing groups. However, the absorption peak at $\sim 1728\text{cm}^{-1}$ is too small to observe, please magnify this peaks.

Our response: The small peaks have been magnified in the Figure.

4. Diameter of bacteriostatic zone in Figure 11 should be measured.

Our response: The diameter of bacteriostatic zone has been measured.

5. Authors should carefully check the grammar throughout the manuscript.

Our response: The grammar of manuscript has been checked.

6. Is there any effect of ascorbic acid concentration and pH of the solution on the size and aggregation of AgNPs on GO?

Our response: In this experiment, neither the concentration of ascorbic acid nor the pH were selected as the orthogonal assay conditions. In the experimental part of the article, the molar ratio of silver nitrate, sodium hydroxide and ascorbic acid is constant. Therefore, this article can not elaborate on the effect of ascorbic acid concentration and pH on the reaction, and we need to further prove it in the following research.

7. Why did the author choose ultrasound assisted AgNPs synthesis instead of microwave-assisted in the experiment?

Our response: Microwave-assisted synthesis of AgNPs is also widely used. Compared with microwave-assisted synthesis AgNPs, the conditions of ultrasonic-assisted synthesis AgNPs are easier to control. Meanwhile, ultrasonic assistance can not only provide the energy required for the reaction, but also prevent the agglomeration of GO. Therefore, the use of ultrasonic to assist the synthesis of silver on the GO surface is more effective than the use of microwave assist.

8. What is the effect of Ag particle size on antibacterial activity?

Our response: In this research, we focused on how to synthesize an effective antibacterial agent that is inexpensive and environmentally friendly. The influence of Ag particle size on antibacterial activity can not be given in detail in the manuscript. In subsequent experiments, we will further study the effect of Ag particle size on antibacterial activity.

All the Changes made in the revised manuscript are highlighted in red.

Thank you very much.

Sincerely,

Haitao Ni